# Fractionated Stereotactic Radiotherapy with Helical Tomotherapy for Brain Metastases: A Mono-Institutional Experience

**DOI:** 10.3390/jpm13071099

**Published:** 2023-07-05

**Authors:** Francesco Cuccia, Salvatore D’Alessandro, Giuseppe Carruba, Vanessa Figlia, Antonio Spera, Daniela Cespuglio, Gianluca Mortellaro, Giuseppina Iacoviello, Antonio Lo Casto, Giovanni Tringali, Giuseppe Craparo, Livio Blasi, Giuseppe Ferrera

**Affiliations:** 1Radiation Oncology, ARNAS Civico Hospital, 90100 Palermo, Italy; vanessa.figlia@arnascivico.it (V.F.); antonio.spera@arnascivico.it (A.S.); daniela.cespuglio@arnascivico.it (D.C.); gianluca.mortellaro@arnascivico.it (G.M.); giuseppe.ferrera@arnascivico.it (G.F.); 2Radiation Oncology School, University of Palermo, 90133 Palermo, Italy; titodales@gmail.com (S.D.); antonio.locasto@unipa.it (A.L.C.); 3Division of Internationalization and Health Research (SIRS), ARNAS Civico Hospital, 90100 Palermo, Italy; giuseppe.carruba@arnascivico.it; 4Medical Physics, ARNAS Civico Hospital, 90100 Palermo, Italy; giuseppina.iacoviello@arnascivico.it; 5Neurosurgery Unit, ARNAS Civico Hospital, 90100 Palermo, Italy; giovanni.tringali@arnascivico.it; 6Neuroradiology Unit, ARNAS Civico Hospital, 90100 Palermo, Italy; giuseppe.craparo@arnascivico.it; 7Medical Oncology, ARNAS Civico Hospital, 90100 Palermo, Italy; livio.blasi@arnascivico.it

**Keywords:** brain metastases, stereotactic radiotherapy, helical tomotherapy

## Abstract

*Background:* The present study reports on the outcomes of our mono-institutional experience of Helical Tomotherapy (HT)-based SRT for brain metastases. The use of this linac is less frequently reported for this kind of treatment. *Methods:* This retrospective study displays a series of patients treated with HT-SRT. The eligibility of using SRT for brain metastases was defined by a Karnofsky performance status of >70, a life expectancy of >6 months, and controlled extra-cranial disease; no SRT was allowed in the case of a number of brain metastases larger than 10. All the cases were discussed by a multidisciplinary board. Toxicity assessments were performed based on CTCAE v5.0. Survival endpoints were assessed using the Kaplan–Meier method, and univariate and multivariate analyses were carried out to identify any potential predictive factor for an improved outcome. *Results:* Sixty-four lesions in 37 patients were treated using HT-SRT with a median total dose of 30 Gy in five fractions. The median follow-up was 7 months, and the 1- and 2-year LC rates were both 92.5%. The IPFS rates were and 56.75% and 51.35%. The OS rates were 54% and 40%. The UA showed better IPFS rates significantly related to male sex (*p* = 0.049), a BED_12_ of ≥42 Gy (*p* = 0.006), and controlled extracranial disease (*p* = 0.03); in the MA, a favorable trend towards LC (*p* = 0.11) and higher BED (*p* = 0.11) schedules maintained a correlation with improved IPFS rates, although statistical significance was not reached. *Conclusions:* HT-based SRT for brain metastases showed safety and efficacy in our monoinstiutional experience. Higher RT doses showed statistical significance for improved outcomes of LC and OS.

## 1. Introduction

Brain metastases are a significant cause of morbidity and mortality in patients with metastatic cancer, with an incidence of up to 65% during the course of illness [1,2]. The most common primary sites are lung, melanoma, renal, breast, and colorectal cancer [3]. Traditionally, options for patients with brain metastases have been limited to whole brain radiotherapy (WBRT) or supportive care alone, and systemic chemotherapy is often discontinued.

While the presence of brain metastases has been classically associated with a poor prognosis, patient outcomes have improved dramatically over the past two decades due to earlier detection [4], improved systemic therapies [5], and the improved management of disease within the brain [6].

As systemic therapies have become more effective in patients with metastatic disease, improved survival rates have recently been seen. Consequently, the management of brain metastases has become a major focus of cancer research, with the intent of improving intra-cranial control and reducing neurological deaths [7].

Although the role of neurosurgery was established in the 1990s as a means for achieving local control and prolonging survival, it was reserved to a small proportion of patients presenting with a single metastasis and no other disease outside the brain [8,9,10].

The increasing number of patients with brain metastases and controlled extracranial disease has led to the search for a non-invasive focal ablative treatment that could be applied efficiently to a much larger population of patients. This set the stage for the development of stereotactic radiotherapy (SRT), a targeted ablative radiation treatment delivered with submillimeter precision to the localized tumor in one fraction (i.e., stereotactic radiosurgery—SRS) or up to five fractions. It is postulated that additional biologic factors or cellular pathways specific to a high dose per fraction of radiation may be involved in the pathophysiology of SRS responses [11].

Moreover, SRT has demonstrated superiority to traditional whole brain radiotherapy (WBRT) with regard to cognitive outcomes in multiple randomized trials [12,13,14]. Once used predominantly for one to three brain metastases [15], SRT usage has expanded to include more numerous [16] and larger brain metastases [17], as defined by the most recent international guidelines [18,19].

While the use of SRT is associated with improved cognition in many patients with brain metastases, patients treated with SRT alone do have a higher rate of developing new brain metastasis in the future [20], a phenomenon named distant brain failure (DBF). Statistical models have recently been published attempting to predict patterns of DBF [21,22].

Recently, several authors have reported that helical tomotherapy (HT) could provide both SRS and SRT. In most studies, the outcomes of SRS/SRT, with or without combined WBRT, have been reported [23,24,25,26,27,28], showing that HT plans, on average, met the general planning objectives for the SRT of multiple lesions in regard to the conformity and homogeneity of the target coverage and sparing of the organs at risk. Compared to other SRS techniques, HT allows for the optimization of dose delivery considering the contribution from all lesions treated through the simultaneous optimized planning and treatment of multiple lesions. In the present study, we report the outcomes of our mono-institutional experience of Helical Tomotherapy (HT)-based SRT for brain metastases.

## 2. Methods

The present study collected the data of a mono-institutional retrospective series of patients treated with fractionated Helical-Tomotherapy-based stereotactic radiotherapy for brain metastases. In our institution, the eligibility to treat brain metastases with SRT was defined by a Karnofsky performance status of >70, a life expectancy of >6 months, and controlled extra-cranial disease; no SRT was allowed in the case of a number of brain metastases larger than 10 (treated otherwise with whole brain radiotherapy plus a simultaneous integrated boost). Concurrent systemic therapy was not an exclusion criterion. All the cases were discussed by a multidisciplinary board with a neuro-radiologist, medical oncologist, neurosurgeon, and radiation oncologist. Written informed consent for each treatment was obtained.

For the patient positioning, a 1.25 mm slice thickness non contrast-enhanced CT simulation scan was acquired in the supine position, using a thermoplastic mask for immobilization.

For planning purposes, image co-registration with a 1 mm contrast-enhanced T1-weighted brain MRI helped with the gross tumor volume (GTV) delineation. The GTV was defined as the entire lesion volume as visualized on the CT/MRI fusion; the planning target volume (PTV) was created by an isotropic 2 mm expansion around the GTV to account for imaging fusion uncertainty, contouring variations, setup errors, and potential patient motion during the treatment. The eyes, lenses, optic nerves, chiasm, brainstem, cord, and the whole brain parenchyma, excluding the PTV and cochlea, were delineated as critical structures. Before each fraction, daily megavoltage tomography was performed for the setup verification.

The prescription dose and fractionation were chosen depending on the lesion size, anatomical site, and its proximity to critical structures; the dose constraints were derived from the peer-reviewed literature [29,30].

The patient details were recorded in a standardized database, which included the personal data of the patients, their treatment information (dose per fraction, total dose, pharmacological therapy and dose, clinical and radiological response, acute, and late side effects), and follow-ups.

After the radiotherapy, physical examinations were set at the first 60 days after the end of the treatment with a contrast-enhanced brain MRI. Subsequent visits with diagnostic imaging were scheduled every 3 months.

The Common Criteria for Adverse Events (CTCAE) v5.0 and Response Evaluation Criteria In Solid Tumors (RECIST V1.1) were used to grade adverse events and evaluate the radiologic tumor response. Local control (LC) was defined as the absence of recurrent disease within the RT field, starting from the end of treatment; Intracranial Progression-Free Survival (IPFS) as the time interval between the RT and the occurrence of new brain metastases; Systemic Progression-Free Survival (SPFS) as the time interval between the RT and the occurrence of any new metastasis; and the Overall Survival (OS) as the time interval between the RT and the death of the patient or the last known follow-up. Acute toxicity was considered as any event occurring within 90 days from the end of treatment, while late toxicity as any event occurring after 90 days from the last session of radiotherapy.

### Statistical Analysis

The baseline characteristics of the patients were collected using descriptive statistics. Survival estimates were performed with the Kaplan–Meier method. Cox regression was applied for the uni- and multi-variate analyses, which were performed assuming a *p*-value of ≤0.05 as statistically significant, including for multivariate analysis *p*-values of <0.15. All the statistical analyses were carried out using the software Graphpad Prism V9.4.1 (Graphpad, San Diego, CA, USA).

## 3. Results

From March 2018 to June 2022, a total of 64 lesions in 37 patients were treated with HT-SRT for a median total dose of 30 Gy (range, 28–30 Gy) with a median number of fractions of five (3–5). The median age of the patients was 59 years (41–80 years); in 86.4% (*n* = 32) of cases, SRT for brain metastases was proposed for patients with controlled extracranial disease.

The most frequent primary histologies were NSCLC, colorectal cancer, and breast cancer, respectively, in 54%, 13.6%, and 16.2% of cases. Concurrent systemic therapy was administered in 56% of patients. The majority of the treated lesions were supratentorial (84%), and cerebellar in the remaining 16%. The median GTV was 0.77 cc (0.3–17.86 cc), and median PTV was 2.66 cc (0.74–30.69 cc). The patients’ characteristics are summarized in Figure 1.

### 3.1. Survival Outcomes

With a median follow-up of 7 months (1–38), the 1- and 2-year local control (LC) rates were both 92.5% (Figure 2).

In the univariate analysis (UA), controlled primary disease and RT doses with a BED_12_ of ≥42 Gy were found to be predictive of improved LC rates, but in the multivariate analysis (MA), only higher doses of radiotherapy maintained statistical significance (*p* = 0.008).

Intracranial progression occurred in 48.6% of cases (*n* = 18), with radiotherapy being proposed as a salvage treatment in seven cases, consisting of further stereotactic treatment in two patients and whole brain radiotherapy in five patients.

The median intracranial progression-free survival (IPFS) was 5 months (1–17 months), with 1- and 2-year rates of 56.75% and 51.35%, respectively. The UA showed better IPFS rates significantly related to male sex (*p* = 0.049), a BED_12_ of ≥42 Gy (*p* = 0.006), and controlled extracranial disease (*p* = 0.03) (Figure 3 and Figure 4).

The systemic progression-free survival (SPFS) rates were, respectively, 51.35% and 48.64% at 1 and 2 years, without any predictive factor for improved outcomes in the MA.

At the time of the analysis, 22 patients died and no patient was lost upon follow-up. The median overall survival (OS) was 7 months (3–38), with 1- and 2-year rates of 54.05% and 40.54%: in the UA, a BED_12_ of ≤42 Gy (*p* = 0.01) and uncontrolled extra-cranial disease (*p* = 0.03) were found to be predictive of worse survival outcomes, but only a BED_12_ of ≤42 Gy maintained statistical significance in the MA (*p* = 0.046).

### 3.2. Toxicity

All patients received brain SRT with concurrent steroid administration. The treatment was well tolerated with no acute or late G ≥ 3 adverse events. Only nausea and fatigue G1 were reported in two patients, who fully recovered within 2 weeks from the end of the RT. During the follow-up, no radiological evidence of radiation necrosis was observed.

## 4. Discussion

The role of stereotactic radiotherapy in the treatment of brain metastases is rapidly increasing as an attractive and effective alternative to conventional whole brain radiotherapy, as a therapeutic option able to provide an improved LC with a minimal neurotoxicity incidence, in a favorable alliance with novel systemic therapies [31].

Several studies are currently available supporting the feasibility and efficacy of this approach, also in cases of patients with a number of metastases of ≥10, traditionally candidates for palliative treatments [32,33,34].

In this scenario, the use of Helical Tomotherapy for brain stereotactic radiosurgery (SRS) has been less reported compared to other techniques such as Gamma- or Cyber-knife or VMAT [35,36].

In a study by Barra et al. [37], SRS was performed with Helical Tomotherapy in single-session treatments with a median dose of 20 Gy (range, 15–20 Gy) for a series of 46 metastases. In this experience, a metal head ring device was applied for patient immobilization, using a 3 mm isotropic margin to generate the PTV from the CTV. The LC rates at 12 months were about 60%, which is lower compared to our 1- and 2-year rates. This might be related to two factors in the abovementioned study: first, the adoption of a dose of 15 Gy in a single fraction for lesions larger than 3 cm, corresponding to a BED_12_ of <40 Gy, which is inadequate for achieving an improved LC; secondly, a higher proportion of patients with a radioresistant histology. In our series, we adopted thermoplastic mask-based patient immobilization and applied a fractionated approach for the delivery of the treatment.

Interestingly, we observed a statistically significant impact of higher doses on improved LC rates, as also reported by Putz et al., who suggested that fractionated schedules may enhance the therapeutic ratio in the treatment of brain metastases, decreasing the incidence of radionecrosis not only for larger lesions, but also in the case of smaller volumes [38].

The favorable impact of fractionated schedules on LC was also reported by Nagai et al., in a series of 128 metastases treated in 54 patients, with the LC, IPFS, and OS rates being comparable with our experience [39].

Similar to the abovementioned study, we adopted the schedule of 28 Gy in four fractions as the most frequently used, obtaining optimal results not only in terms of clinical outcomes, but also in terms of adverse events.

Multifraction SRT treatments were also recently reported by Kornhuber et al. [40] from a physicist perspective, with the aim of delivering a more homogeneous dose to the GTV, with a simultaneous integrated boost (SIB) approach. Starting from a five-fraction prescription with a total dose of 35 Gy to the PTV, the authors suggested the potential favorable impact of this approach as a means to reducing the risk of radiation necrosis.

In agreement with these results, this finds confirmation in a recent publication by Di Perri et al. [41], who analyzed the patterns of radiation necrosis occurrence in a series of 360 lesions treated in 294 patients who received fractionated SRT. Interestingly, lower BED schedules were related to poorer outcomes in terms of the LC for intact metastases, while no difference was observed for adjuvant SRT treatments. Regarding the incidence of radiation necrosis, a higher risk was detected with higher BED schedules and in the case of concomitant immunotherapy within 3 months of the SRT delivery.

Another retrospective study was recently published by Layer et al. [42], reporting the outcomes of 49 metastases treated in 36 patients with a fractionation schedule of 35 Gy in five fractions, with a 1-year LC rate of 83% and also a radiation necrosis rate of 14%. The higher cumulative dose of this series might explain the higher incidence of radiation necrosis, compared to our series, where no radiation necrosis was detected after treatment. Notably, this study also included adjuvant surgical bed stereotactic treatments and a large proportion of patients (about 25%) who received whole brain radiotherapy prior to SRT.

Concerning LC rates, the authors addressed the suboptimal result of the presence of radioresistant histologies in their series, such as melanoma, overrepresented in comparison to other primary tumors, such as breast cancer.

In agreement with the abovementioned studies, in our series, better LC rates were also found to be predictive of improved intracranial progression-free survival, although this did not reach statistical significance (*p* = 0.11); nonetheless, this evidence supports the role of local treatments as an alternative strategy to delaying the administration of salvage whole brain radiotherapy or the initiation of new systemic agents.

This also finds confirmation in a recent experience published by Soni et al. [43], in which the incidence of local failure was found to be predictive of a higher use of salvage whole brain radiotherapy.

Notably, in our series, only two patients received a second SRT for the occurrence of newly diagnosed brain metastases after the first treatment. However, the versatility of this technique in postponing, as much as possible, the administration of whole brain radiotherapy has been reported by several experiences, with the advantage of reducing the detrimental impact on neurocognition [13].

Our favorable outcomes in terms of the LC might also be related to the limited presence of radioresistant histologies in our cohort, with ovarian cancer, melanoma, and renal cell carcinoma consisting of only 5 cases out of 37.

A recent study by Gruber et al. [44] reported the outcomes of a cohort of 73 patients with 103 brain metastases treated with a fractionated schedule in six sessions (30 Gy using 5 Gy per fraction). The series also included radioresistant histologies, with melanoma being represented in 34.2% of cases, collecting global local progression-free survival rates of 68.7% and 61.6% at 12 and 24 months. These results are consistently lower than our 1- and 2-year LC rates, likely due to two main factors: first, the greater presence of radioresistant histologies in the above-mentioned study, and secondly, the higher BED of our prescription regimens.

Concerning the incidence of radiation necrosis, we did not observe any case of radiation necrosis, likely due to the small size of our cohort. Another factor likely related to these data is the relatively small volume of the GTV in our series.

This finds confirmation in a recent publication by Johannwerner et al. [45], with a study cohort of 218 brain lesions in 169 patients treated with single-fraction radiosurgery or fractionated stereotactic radiotherapy. The authors reported that single-fraction schedules were significantly associated with a higher risk of radiation necrosis, especially for lesions sized more than 20 mm, while, on the contrary, for fractionated regimens, the risk of radiation necrosis was lower, regardless of the lesion size.

One of the largest experiences addressing the risk of radiation necrosis for stereotactic radiotherapy for large (≥2.0 cm) brain metastases was published by Minniti et al. [46]

In this series, 289 patients were treated either with radiosurgery or fractionated stereotactic radiotherapy; the authors reported improved LC rates and a reduced risk of radiation necrosis in the cohort of patients treated with fractionated regimens (usually 27 Gy in three fractions) compared to those treated with single-fraction schedules, recording significantly higher brain V12 Gy volumes in the radiosurgery cohort compared to the multifraction subgroup.

In our series, no constraint violations were reported for all the treatment plans, and no more than four metastases were simultaneously treated, leading to a lower low-dose bath with a consequently reduced risk of radiation necrosis.

Our study has several limitations: at first, the small sample size and retrospective nature of the study affected the statistical power of the results. Secondly, the relatively short follow-up may have limited definitive conclusions about long-term outcomes, but the natural history of oncological patients with brain metastases must be taken into account. Moreover, different primary histologies were included, making it hard to draw definitive conclusions about the impact of this treatment on the natural history of the disease; also, ongoing systemic therapies may have contributed to the LC rates, although they were not statistically significantly related, probably due to the small sample size.

Nonetheless, this study provides further evidence in support of Helical Tomotherapy for brain SRT, highlighting the favorable impact of fractionated schedules on the delivery of the treatment.

Further evidence in support of an optimal fractionation schedule for the stereotactic treatment of brain metastases will be provided by the FSRT-trial [47], a randomized phase III trial comparing fractionated schedules (12 Gy × four fractions to the 80% isodose encompassing the PTV) vs. SRS according to RTOG 9005 for brain metastases ranging between 1 cm and 4 cm, with randomization based on primary histology and lesion size. The primary endpoint of the study is the time to local progression; toxicity, quality of life, and overall survival are among the secondary endpoints.

## 5. Conclusions

This mono-institutional experience supports the use of Helical Tomotherapy for brain metastases fractionated stereotactic radiotherapy as a safe and effective treatment option, with preliminary encouraging results in terms of its clinical outcomes and no incidence of severe adverse events.

## Figures and Tables

**Figure 1 jpm-13-01099-f001:**
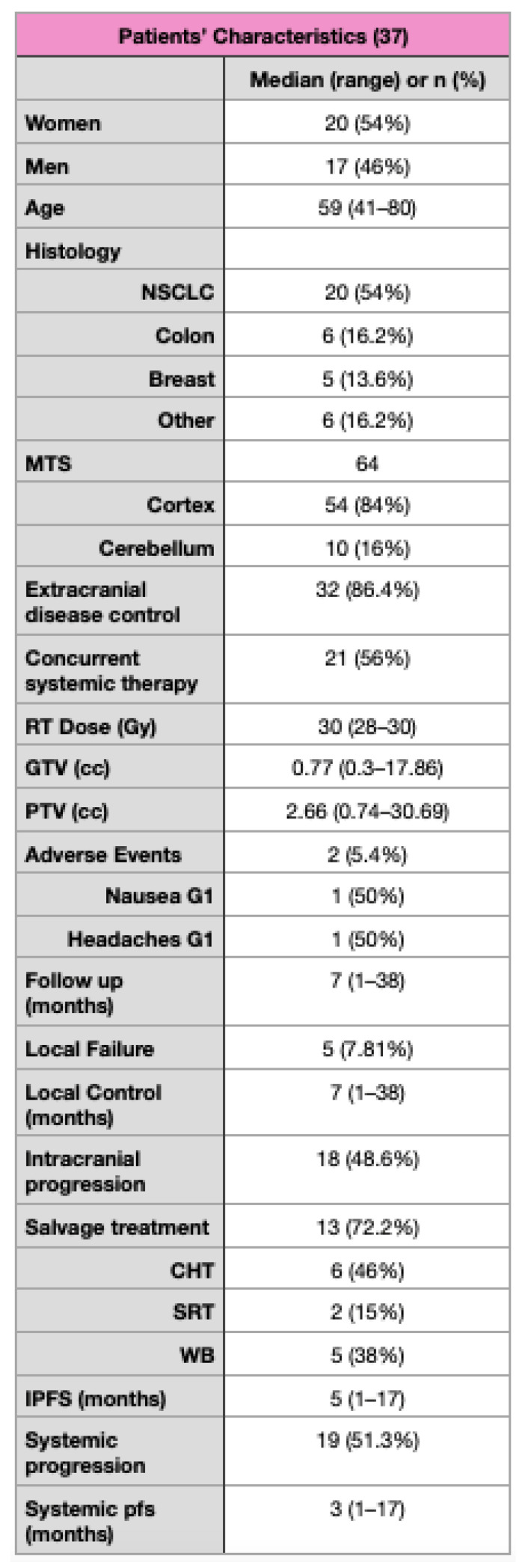
Patients’ characteristics.

**Figure 2 jpm-13-01099-f002:**
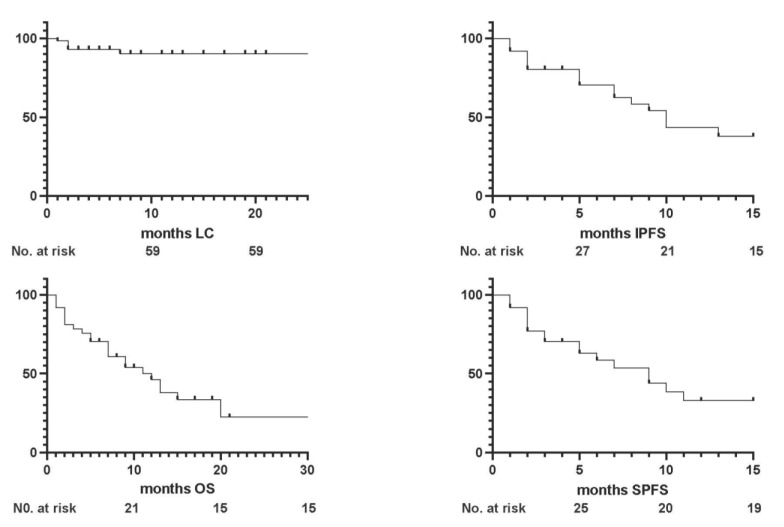
Survival curves for clinical outcomes.

**Figure 3 jpm-13-01099-f003:**
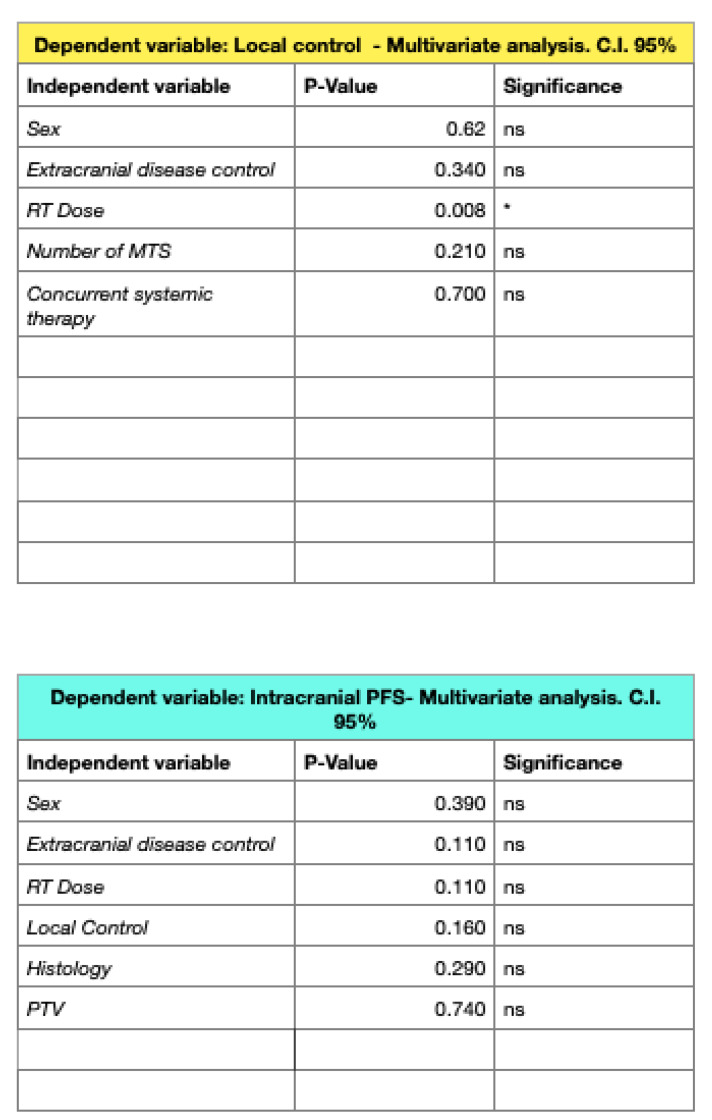
Multivariate analyses for LC and IPFS. * *p*-Value < 0.05.

**Figure 4 jpm-13-01099-f004:**
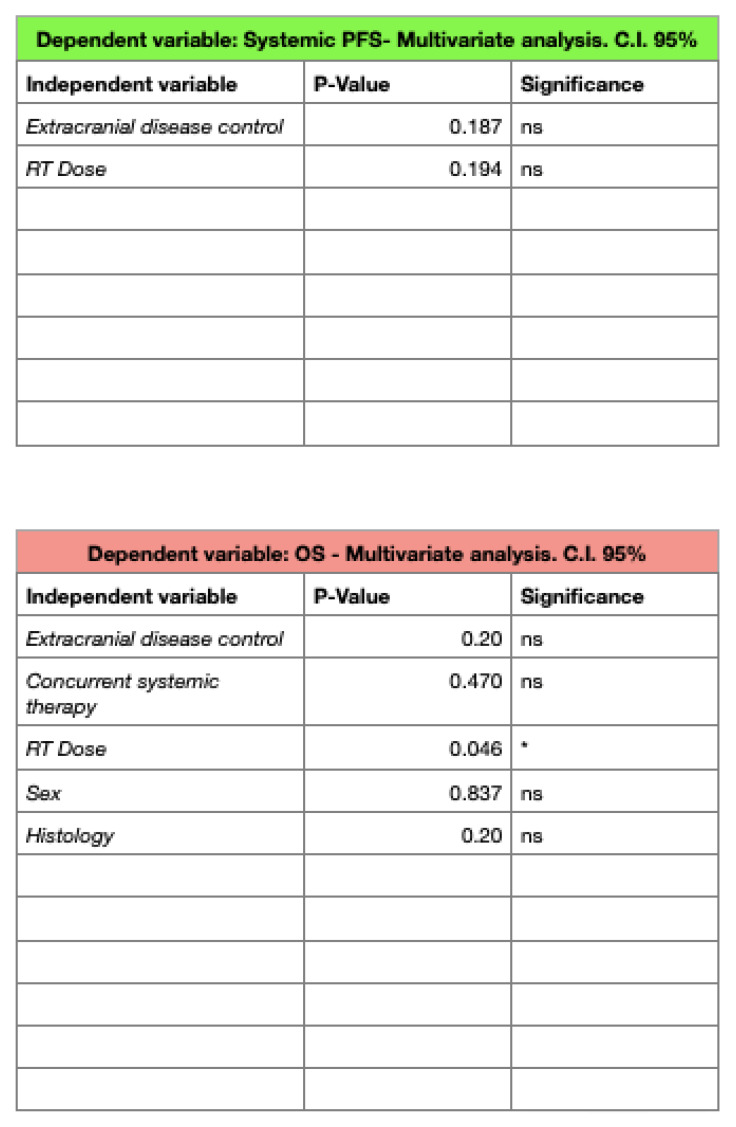
Multivariate analyses for SPFS and OS. * *p*-Value < 0.05.

## Data Availability

Data availability at authors’ discretion.

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
