# Peer review of "Fractionated Stereotactic Radiotherapy with Helical Tomotherapy for Brain Metastases: A Mono-Institutional Experience"

_jpm, 2023, doi:10.3390/jpm13071099_

Round 1
Reviewer 1 Report
Authors should be commended on contributing to the treatment of TomoTherapy in patients with brain metastases. The data on treatment outcomes using TomoTherapy is limited in the field of stereotactic radiosurgery for this patient cohort. However, the quality of the findings is hampered by the small patient number and heterogeneous patient characteristics.
This cohort of patients was treated between 2018 and 2022. Systemic therapy including targeted therapy and immunotherapy is supposed to be administered, which dramatically improves the local tumor control and overall survival. Those important data were not reflected in the analysis.
In the current study, the radiation dose range varied from 28 to 30 Gy in 3 to 5 fractions. Is this considered over-treated in a tumor size of 0.03cc?
In Page 8, it shows a follow-up of 7 months (3 – 38) which does not match the one in the figure 1.
How many patients died at the time of analysis? How many patients were lost to follow-up? In statistics, no such thing is called a favorable trend toward LC (p=0.11).
What is the definition of local tumor control, systemic progression-free survival, and overall survival? I assume that this is the actuarial tumor control rate. If so, please state it clearly. In the Kaplan-Meier plot, it is important to present the patient number at risk underneath the x-axis. The value BED12 >= or <= 42Gy is an arbitral number or based on some kind of statistical analysis, which requires clarification. In the statistical analysis section, uni- and multi-variate analyses were performed assuming a p-value…, what statistical model was used to calculate those?
In Figure 4, it is inappropriate to use the parameter of local control in the multivariate analysis of SPFS and OS. Again, what statistical method was used to calculate p values in Figure 4?
Please provide references for the statement in the Method section that “lesion size, anatomical site and proximity to critical structures, dose constraints were derived from peer-reviewed literature”.
improvements are required.
Author Response
Reviewer1 (R1):
Authors should be commended on contributing to the treatment of TomoTherapy in patients with brain metastases. The data on treatment outcomes using TomoTherapy is limited in the field of stereotactic radiosurgery for this patient cohort.
However, the quality of the findings is hampered by the small patient number and heterogeneous patient characteristics. This cohort of patients was treated between 2018 and 2022. Systemic therapy including targeted therapy and immunotherapy is supposed to be administered, which dramatically improves the local tumor control and overall survival. Those important data were not reflected in the analysis.
Authors' Reply (AR): we thank the reviewer for the helpful suggestions. We have highlighted at the end of the Discussion the limitations of the study, reinforcing the inclusion of different histologies and concurrent systemic therapies
R1: In the current study, the radiation dose range varied from 28 to 30 Gy in 3 to 5 fractions. Is this considered over-treated in a tumor size of 0.03cc?
AR: we apologize for the mistake, tumor size range has been corrected to 0.3 cc
R1: In Page 8, it shows a follow-up of 7 months (3 – 38) which does not match the one in the figure 1.
AR: edited and matched to Figure 1
R1: How many patients died at the time of analysis? How many patients were lost to follow-up? In statistics, no such thing is called a favorable trend toward LC (p=0.11).
AR: we have removed the entire sentence concerning the favorable trend toward LC. At time of the analysis 22 patients died; no patients were lost to follow up
R1: What is the definition of local tumor control, systemic progression-free survival, and overall survival? I assume that this is the actuarial tumor control rate. If so, please state it clearly. In the Kaplan-Meier plot, it is important to present the patient number at risk underneath the x-axis.
AR: In the Methods section we have provided definitions for all the clinical outcomes. We have added the number at risk row under each graph.
R1: The value BED12 >= or <= 42Gy is an arbitral number or based on some kind of statistical analysis, which requires clarification. In the statistical analysis section, uni- and multi-variate analyses were performed assuming a p-value…, what statistical model was used to calculate those? In Figure 4, it is inappropriate to use the parameter of local control in the multivariate analysis of SPFS and OS. Again, what statistical method was used to calculate p values in Figure 4?
AR: in the Methods section, we have defined Cox regression analysis for uni- and multi-variate analyses. The 42 Gy BED12 cut-off was proposed since it is considered as the recommended dose for improved LC rates, as reported by the following reference (added to the text): Wiggenraad R, Kanter AV, De, Kal HB, Taphoorn M, Vissers T, Struikmans H. Dose-effect relation in stereotactic radiotherapy for brain metastases. A systematic review. Radiother Oncol. 2011;98(3):292–7.https://doi.org/10.1016/J.RADONC.2011.01.011 . We have removed LC from the multivariate analysis for SPFS and OS
R1: Please provide references for the statement in the Method section that “lesion size,
anatomical site and proximity to critical structures, dose constraints were derived from
peer-reviewed literature”.
AR: we have included a reference to this sentence: “Gérard M, Jumeau R, Pichon B, Biau J, Blais E, Horion J, Noël G. Contraintes de dose en radiothérapie conformationnelle fractionnée et en radiothérapie stéréotaxique dans les hippocampes, le tronc cérébral et l’encéphale : limites et perspectives [Hippocampus, brainstem and brain dose-volume constraints for fractionated 3-D radiotherapy and for stereotactic radiation therapy: Limits and perspectives]. Cancer Radiother. 2017 Oct;21(6-7):636-647. French. Doi:10.1016/j.canrad.2017.08.108.

Reviewer 2 Report
I am grateful for the oppurtunity to review the manuskript jpm-2445432. The authors present a well srtuctured retrospective study of a single instutute cohort of stereotactic RT of brain mets. I recommend to publish the paper.
I have only a few comments/questions:
In the introduction you describe your technique (tomotherapy) as favourable for larger mets. Did you perform a subgroup analysis for these patients (e.g. diamater >20mm)?
You state theat BED12≥42Gy is favourable. Your fractionation schedule of 5x6Gy equals 45Gy. Could you please elaborate, how many patients were radiated on with less than 42Gy BED12, and why? It would be nice to let the reader understand the impact of this finding (i.e. the 42Gy BED12-threshold).
Author Response
Reviewer2: I am grateful for the oppurtunity to review the manuskript jpm-2445432. The authors present a well srtuctured retrospective study of a single instutute cohort of stereotactic RT of brain mets. I recommend to publish the paper.
AR: we thank the reviewer for the helpful comments.
R2:In the introduction you describe your technique (tomotherapy) as favourable for larger mets. Did you perform a subgroup analysis for these patients (e.g. diamater >20mm)?
AR: no subgroup analysis for larger metastases was performed since in our series a small proportion of patients was treated with metastases larger than 20 mm
R2: You state theat BED12≥42Gy is favourable. Your fractionation schedule of 5x6Gy equals 45Gy. Could you please elaborate, how many patients were radiated on with less than 42Gy BED12, and why? It would be nice to let the reader understand the impact of this finding (i.e. the 42Gy BED12-threshold).
AR: The 42Gy BED12 threshold is derived from peer reviewed literature that recommends a minimum 42 Gy dose to achieve adequate local control for brain metastases, as reported by the following reference (added to the text): Wiggenraad R, Kanter AV, De, Kal HB, Taphoorn M, Vissers T, Struikmans H. Dose-effect relation in stereotactic radiotherapy for brain metastases. A systematic review. Radiother Oncol. 2011;98(3):292–7.https://doi.org/10.1016/J.RADONC.2011.01.011
Reviewer 3 Report
The text provides an overview of treating brain metastases using stereotactic radiotherapy (SRT) with Helical Tomotherapy (HT). While the information is informative, there are some areas where the text could be improved:
1. The Introduction section is too long and covers a wide range of information, but the flow of ideas could be improved. It would be helpful to reorganize the paragraphs to present the information in a more logical sequence, ensuring that the reader can follow the development of ideas smoothly.
2.“Compared to other SRS techniques, such as Gamma Knife stereotactic radiosurgery (GKSRS), HT allows the optimization of dose delivery considering the contribution from all lesions treated through the simultaneous optimized planning and treatment of multiple lesions, allowing for the treatment of larger lesions than other SRT techniques GKSRS.” This is outdated information for new-generation Gamma Knife devices with the Lightning planning system. Besides, what do the authors refer to with “larger lesions”? Hypofractionation is also available with Gamma Knife Icon, so we can now also treat larger lesions.
3. The eligibility criteria for brain metastases SRT is briefly mentioned but lacks a detailed explanation. It would be beneficial to provide more information regarding the specific criteria for patient selection, such as excluding patients with more than ten brain metastases. This would help readers understand the study population and its limitations.
4. The authors added a 2 mm expansion around the GTV to account for imaging fusion uncertainty, contouring variations, setup errors, and potential patient motion during treatment. However, the primary disadvantage to this approach is the increased planning target volume, which for most brain metastases may be normal tissue that might otherwise be spared. It is also difficult to predict the appropriate margin, given the differences in the magnitude of MR distortion between each patient. Adding margins (of the same nominal size) to small structures yields differences up to 40%, resulting in substantially inconsistent total volumes. Assuming the same prescription dose is used for both targets, the dose to the normal brain will increase significantly, and, in turn, the risk of radiation necrosis will increase as well. This is particularly important when large numbers of lesions in proximity to each other are treated in the same session because of dose-interplay effects. Do the authors use any distortion correction?
5. As 4-day fractionation is not standard, how did the authors decide which patients to use it?
6. What was the rationale behind the hypofractionated treatment of lesions as small as 0.03 cc?
7. The median follow-up duration is mentioned. Still, it is essential to specify whether the follow-up time refers to the overall survival or the specific outcomes mentioned (e.g., local control, intracranial progression-free survival).
8. The authors mention the ongoing FSRT trial, but providing more information about the trial design, objectives, and expected contributions to the field would be helpful.
9. The text briefly mentions other studies that have explored SRT for brain metastases but does not provide a comprehensive discussion or comparison with the current study. Including a more detailed discussion of previous research and how the present study adds to the existing knowledge would enhance the text.
10. The text includes several references to support the information provided, but the citation style is inconsistent. Following a specific citation style throughout the text would be better for better clarity and consistency.
Author Response
Reviewer3: The text provides an overview of treating brain metastases using stereotactic radiotherapy (SRT) with Helical Tomotherapy (HT). While the information is informative, there are some areas where the text could be improved.
AR: thank you for your helpful comments in order to improve the quality of our manuscript
R3: The Introduction section is too long and covers a wide range of information, but the flow of ideas could be improved. It would be helpful to reorganize the paragraphs to present the information in a more logical sequence, ensuring that the reader can follow the development of ideas smoothly.
AR: the Introduction has been significantly shortened to increase readability
R3:“Compared to other SRS techniques, such as Gamma Knife stereotactic radiosurgery (GKSRS), HT allows the optimization of dose delivery considering the contribution from all lesions treated through the simultaneous optimized planning and treatment of multiple lesions, allowing for the treatment of larger lesions than other SRT techniques GKSRS.” This is outdated information for new-generation Gamma Knife devices with the Lightning planning system. Besides, what do the authors refer to with “larger lesions”? Hypofractionation is also available with Gamma Knife Icon, so we can now also treat larger lesions.
AR: we have removed all the sentences referring to the GK system.
R3: The eligibility criteria for brain metastases SRT is briefly mentioned but lacks a detailed explanation. It would be beneficial to provide more information regarding the specific criteria for patient selection, such as excluding patients with more than ten brain metastases. This would help readers understand the study population and its limitations.
AR: we have explicitly stated that patients with more than 10 metastases were usually treated with whole brain radiotherapy plus simultaneous integrated boost. Also, we have highlighted that concurrent systemic therapy was not an exclusion criterion.
R3: The authors added a 2 mm expansion around the GTV to account for imaging fusion uncertainty, contouring variations, setup errors, and potential patient motion during treatment. However, the primary disadvantage to this approach is the increased planning target volume, which for most brain metastases may be normal tissue that might otherwise be spared. It is also difficult to predict the appropriate margin, given the differences in the magnitude of MR distortion between each patient. Adding margins (of the same nominal size) to small structures yields differences up to 40%, resulting in substantially inconsistent total volumes. Assuming the same prescription dose is used for both targets, the dose to the normal brain will increase significantly, and, in turn, the risk of radiation necrosis will increase as well. This is particularly important when large numbers of lesions in proximity to each other are treated in the same session because of dose-interplay effects. Do the authors use any distortion correction?
AR: no distortion corrections were applied for MRI fusion. The choice of a 2 mm isotropic margin relies on the fact that, unlike dedicated devices such as GammaKnife or Cyberknife, we considered as safe and effective a margin higher than 1 mm. Moreover, 2 mm isotropic margin is also reported in other literature experiences included in the reference list.
R3: As 4-day fractionation is not standard, how did the authors decide which patients to use it?
AR: the 4-day fractionation was the first option at the beginning of our insitutional practice of brain stereotactic radiotherapy, based on a previous helical tomotherapy experience mentioned in the reference list (n.38, Nagai et al.).
R3: What was the rationale behind the hypofractionated treatment of lesions as small as 0.03 cc?
AR: we apologize for the mistake, 0.03 was corrected to 0.3 cc
R3: The median follow-up duration is mentioned. Still, it is essential to specify whether the follow-up time refers to the overall survival or the specific outcomes mentioned (e.g., local control, intracranial progression-free survival).
AR: in the Methods section, we have provided a definition for all the clinical outcomes, assumed from the end of radiotherapy.
R3: The authors mention the ongoing FSRT trial, but providing more information about the trial design, objectives, and expected contributions to the field would be helpful.
AR: we have provided more details about the design and the endpoints of the FSRT trial and also edited the reference
R3: The text briefly mentions other studies that have explored SRT for brain metastases but does not provide a comprehensive discussion or comparison with the current study. Including a more detailed discussion of previous research and how the present study adds to the existing knowledge would enhance the text.
AR: we have included a recent article of fractionated stereotactic radiotherapy and extended the Discussion about the role of fractionated treatments and potential pros and cons of this choice. In the Discussion we have stated the importance of this study in support of Helical Tomotherapy- based stereotactic radiotherapy for brain metastases, in light of the paucity of literature in this scenario.
R3: The text includes several references to support the information provided, but the citation style is inconsistent. Following a specific citation style throughout the text would be better for better clarity and consistency.
AR: reference list is now more consistent. Thank you for your helpful suggestions.